# Predictors for estimating subcortical EEG responses to continuous speech

**Joshua P. Kulasingham** [1]*, **Florine L. Bachmann** [2], **Kasper Eskelund** [3], **Martin Enqvist**[1], **Hamish Innes-Brown**[2,4‡], **Emina Alickovic**[1,2‡]

**1** Automatic Control, Department of Electrical Engineering, Linköping University, Linköping, Sweden, **2** Eriksholm Research Centre, Snekkersten, Denmark, **3** Oticon A/S, Smørum, Denmark, **4** Department of Health Technology, Technical University of Denmark, Lyngby, Denmark

‡ HIB and EA are contributed equally to this work and considered as senior authors.
* joshua.kulasingham@liu.se

**Data Availability Statement:** There are ethical restrictions on sharing the data set. The consent given by participants at the outset of this study did not explicitly detail sharing of the data in any

## Abstract

Perception of sounds and speech involves structures in the auditory brainstem that rapidly process ongoing auditory stimuli. The role of these structures in speech processing can be investigated by measuring their electrical activity using scalp-mounted electrodes. However, typical analysis methods involve averaging neural responses to many short repetitive stimuli that bear little relevance to daily listening environments. Recently, subcortical responses to more ecologically relevant continuous speech were detected using linear encoding models. These methods estimate the temporal response function (TRF), which is a regression model that minimises the error between the measured neural signal and a predictor derived from the stimulus. Using predictors that model the highly non-linear peripheral auditory system may improve linear TRF estimation accuracy and peak detection. Here, we compare predictors from both simple and complex peripheral auditory models for estimating brainstem TRFs on electroencephalography (EEG) data from 24 participants listening to continuous speech. We also investigate the data length required for estimating subcortical TRFs, and find that around 12 minutes of data is sufficient for clear wave V peaks (>3 dB SNR) to be seen in nearly all participants. Interestingly, predictors derived from simple filterbank-based models of the peripheral auditory system yield TRF wave V peak SNRs that are not significantly different from those estimated using a complex model of the auditory nerve, provided that the nonlinear effects of adaptation in the auditory system are appropriately modelled. Crucially, computing predictors from these simpler models is more than 50 times faster compared to the complex model. This work paves the way for efficient modelling and detection of subcortical processing of continuous speech, which may lead to improved diagnosis metrics for hearing impairment and assistive hearing technology.

## Introduction

The human auditory system consists of several subcortical and cortical structures that rapidly process incoming sound signals such as speech. Electroencephalography (EEG) measurements of the aggregate activity of these neural structures have been instrumental in understanding

format; this limitation is keeping with EU General Data Protection Regulation, and is imposed by the Research Ethics Committees of the Capital Region of Denmark. Due to this regulation and the way data was collected with a low number of participants, it is not possible to fully anonymize the dataset and hence cannot be shared. As a non-author contact point, data requests can be sent to Claus Nielsen, Eriksholm research operations manager at clni@eriksholm.com.

**Funding:** JPK has received funding from the William Demant Foundation (Case no. 20-0480). Oticon A/S provided support in the form of salaries for authors FLB, KE, HI, EA, but did not have any additional role in the study design, data collection and analysis, decision to publish, or preparation of the manuscript.

**Competing interests:** The authors have declared that no competing interests exist. The commercial affiliation of authors FLB, KE, HI and EA does not alter our adherence to PLOS ONE policies on sharing data and materials.

the mechanisms underlying normal hearing and hearing impairments [1, 2]. One important measure is the morphology of the auditory brainstem response (ABR), and the amplitude and latency of ABR peaks have been widely used in many clinical settings such as neonatal hearing screening [3]. Conventional methods to detect the ABR rely on averaging responses over multiple trials of non-natural, short stimuli such as clicks, chirps or speech syllables [4].

Recently, ABR-like responses to continuous, ongoing speech were detected [5, 6], allowing for the exploration of subcortical processing of ecologically relevant speech stimuli. One method to estimate these subcortical responses is the temporal response function (TRF), a linear encoding model of time-locked neural responses to continuous stimuli [7]. TRFs have been widely used for estimating *cortical* responses to speech [8–12], but fewer studies have investigated *subcortical* TRFs [5, 13–16].

Several factors complicate the direct application of TRF models to detect subcortical responses to continuous speech. Electrical responses that are generated in the brainstem and measured at the scalp are small compared to the amplitude of the on-going EEG. They are subsequently difficult to detect, and a large amount of data is required for reliable TRF estimation [5, 13]. Additionally, subcortical neural processes rapidly time-lock to fast stimulus fluctuations, and a measurement system with precise synchronization (sub-millisecond) between the stimulus and the EEG is essential in order to extract these responses. Another concern is that linear models may ignore several highly non-linear and adaptive processing stages in the auditory periphery and brainstem [17–19]. The TRF is a linear model that relates the EEG signal to a stimulus-derived predictor, and therefore cannot capture the non-linear processing stages of the auditory system. However, the predictor, serving as the input to the TRF model, can be constructed to be a feature (or transformation) of the speech stimulus relevant to the auditory system. Accounting for peripheral non-linearities in the predictor could help 'linearize' the TRF estimation problem and lead to improved TRF models that reflect the activity of later neural processes.

Previous work has used the rectified speech waveform as a predictor, which is a coarse approximation of the initial rectifying non-linearity in the cochlea [5]. However, a recent study has shown that predictors derived from a complex model of the auditory periphery [20] that incorporates non-linear stages can lead to improved subcortical TRFs [21]. Another recent study showed that auditory-model-derived predictors outperform previously used envelope predictors even for *cortical* TRFs [22].

It is essential to determine computationally efficient predictors that result in clear TRF peaks for clinical applications involving realistic speech stimuli, or for future assistive hearing technology. Previous work has compared different methods to compute envelope predictors for investigating *cortical* responses to continuous speech [23]. In this work, we compared the rectified speech waveform with predictors derived from various auditory models in terms of their suitability for estimating subcortical TRFs. We computed predictors from filterbank models [17], with or without adaptation [24], and compared them to a more complex auditory nerve model [20, 25] that has been previously used to fit subcortical TRFs [21]. TRFs were estimated from EEG data recorded from 24 participants listening to continuous speech. Prior work indicates that the most prominent feature of subcortical TRFs is the wave V peak [5, 15], which was used as a performance measure in our study. Additional measures such as the computational time taken to generate predictors and the amount of data required for fitting TRFs for each predictor type are also reported.

We corroborate recent findings [21] by confirming that the predictor derived from a complex model [20] of the auditory nerve outperforms the rectified speech predictor. Interestingly, our results indicate that predictors from simpler models [24] can reach similar performance for estimating wave V peaks as complex models, with the added advantage of being more than

50 times faster to compute. These simpler models, combined with TRF analysis, could lead to efficient algorithms for future assistive hearing technology [26], and encourage the use of more ecologically relevant continuous speech stimuli in clinical applications.

## Materials and methods

### Experimental setup

EEG data was collected from 24 participants with clinically normal hearing thresholds (14 males, $M_{age}$ = 37.07, $SD_{age}$ = 10.02 years). All participants provided written informed consent and the study was approved by the ethics committee for the capital region of Denmark (journal number 22010204). The data collection period was from 1st June to 11th October 2022. EEG data was recorded while participants were seated listening to continuous segments from a Danish audiobook of H.C. Andersen adventures, read by Jens Okking. The participants were instructed to relax and listen to the story. Four audiobook stories were presented in randomized order, divided into two segments ($M_{duration}$ = 6 minutes 0 seconds, $SD_{duration}$ = 55 seconds) each, resulting in a total of 8 trials.

The 2-channel audio was averaged to form a mono audio channel, which was then highpass filtered at 1kHz using a first order Butterworth filter to enhance the relative contribution of high frequencies, since the brainstem response is more strongly driven by high frequencies [27]. Using this gentle highpass filter resulted in natural sounding speech in which a lot of power between 125-1000 Hz as well as the pitch information is clearly preserved. This method was also used in prior studies to detect clear subcortical TRFs [5]. The single channel speech segments were calibrated to be 72 dB SPL using the following procedure: Speech shaped noise was generated by transforming white noise to have the long-term spectrum of the speech. This signal was then calibrated to be 72 dB SPL by recording the audio signals using a measurement amplifier (Bruel and Kjær Type 2636) and head-and-torso simulator (HATS, Bruel and Kjær Type 4128-C) containing two ear simulators (Bruel and Kjær Type 4158). The setup was calibrated using a sound source (Bruel and Kjær Type 4231). Each speech segment was then scaled digitally to have the same root mean square (r.m.s.) value as the 72 dB SPL speech shaped noise. These speech signals were then presented binaurally using an RME Fireface UCX soundcard (RME Audio, Haimhausen Germany) and Etymotic ER-2 (Etymotic Research, Illinois, USA) insert earphones, which were shielded using a grounded metal box to avoid direct stimulus artifacts on the EEG. Stimulus artifacts occur when electromagnetic activity related to stimulus presentation is recorded in the EEG, and are largely caused by electromagnetic leakage of the headphone transducers and cables [28]. Here, we employed several methods to reduce stimulus artifacts: 1) Air-tube insert earphones were employed, creating distance between the headphone transducers and the EEG electrodes. 2) The headphone transducers were shielded with grounded metal boxes which has been shown to reduce stimulus presentation artifacts [28, 29]. The audio signal cables were also shielded, with the cable shield connected to the same ground as the metal box. 3) Model predictors were computed once for the original and once for the sign-inverted speech stimuli. TRFs were computed for both predictors, and then averaged, following prior work [5]. This approach is inspired by the traditional approach of using repeated short stimuli of alternating polarity, and then averaging across neural responses. Further details on predictor and TRF estimation are provided below. Later visual inspection of TRFs confirmed that stimulus artifacts were not present in the estimated TRFs.

### EEG data collection and preprocessing

A Biosemi 32-channel EEG system was used with a sampling frequency of 16,384 Hz and a fifth order cascaded integrator-comb anti-aliasing filter with a -3 dB point at 3276.8 Hz.

Electrodes were placed on the mastoids and earlobes, as well as above and below the right eye. Scalp electrodes were placed according to the 10-20 system. Data analysis was conducted in MATLAB (version R2021a) and the Eelbrain Python toolbox (version 0.38.1) [30] using only the Cz electrode referenced to the average of the two mastoid electrodes. The EEG data was highpass filtered using a first order Butterworth filter with cutoff frequency of 1 Hz. To remove power line noise, the signal was passed through FIR notch filters at all multiples of 50 Hz until 1000 Hz, with widths of 5 Hz. The data was then downsampled to 4096 Hz (after passing through an anti-aliasing lowpass filter) to speed up computation. Simple artifact removal was performed by zeroing out 1 second segments around parts of the EEG data that had amplitudes larger than 5 standard deviations above the mean, similar to prior work [5]. Finally, only the data from 2 to 242 seconds of each trial was used for further analysis to avoid onset effects and to have the same amount of data in each trial.

Detecting subcortical responses requires precise synchronization between the EEG and the audio stimuli. Hence, to avoid trigger jitters and clock drifts, the output of the audio interface was also fed to the BioSemi Erg1 channel via an optical isolator to maintain electrical separation between the mains power and the data collection system (StrimTrak, BrainProducts GmbH, Gilching, Germany). The recorded signal from the StimTrak was used to generate predictors for the TRF analysis.

## Auditory models

Predictors were computed using several auditory models, described below in order of increasing complexity. For all models, the input was the audio stimulus, as recorded by the StrimTrak system. The lags inherent in the output of each model were accounted for by shifting the generated predictors to maximize the correlation with the rectified speech predictor. Since brainstem responses are largely agnostic to stimulus polarities, a pair of predictors were generated for each model, using an input stimulus pair with the original stimulus and the stimulus with opposite sign. In line with prior work [5, 21], TRFs were fit to both predictors separately and then averaged together. Although some auditory models account for peripheral rectification, predictor pairs were generated for all models in order to have the same preprocessing setup and since averaging the two TRFs fit to the predictor pair led to cleaner estimates. The generated predictor waveforms are shown in Fig 1 for a short speech segment, and the overall

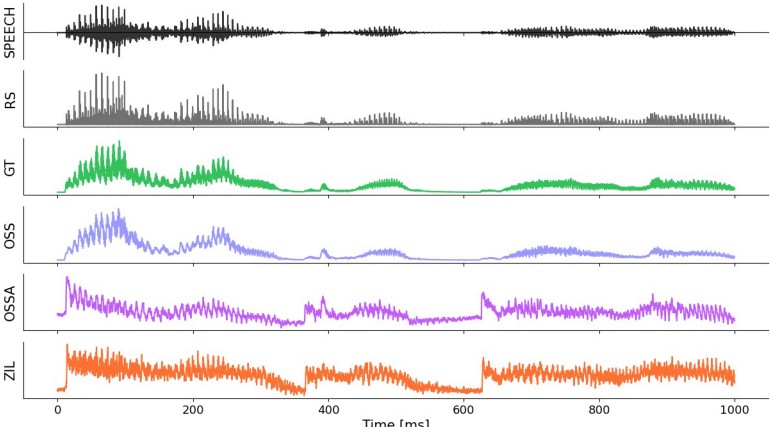

**Fig 1. Predictor waveforms.** The predictor waveforms are shown for a 1-second speech segment (also shown in the top row) to illustrate the differences between the models. Note that the OSSA and ZIL predictors are the most different from the speech waveform, since they incorporate more peripheral non-linearities and adaptation effects.

**Table 1. Predictor comparison.** The ZIL model is more than 50 times slower to compute than the other models.

| Predictor | Computation Time (1 s input) | Correlation with RS | Correlation with ZIL |
|---|---|---|---|
| RS | - | - | 0.316 |
| GT | 0.0521 s | 0.461 | 0.550 |
| OSS | 0.0563 s | 0.438 | 0.496 |
| OSSA | 0.0680 s | 0.262 | 0.577 |
| ZIL | 4.1208 s | 0.316 | - |

execution times to generate each predictor and the Pearson correlations between predictors are also reported in Table 1.

**Rectified speech (RS).** Previous studies have shown that the rectified speech signal can be used to estimate subcortical TRFs to continuous speech [5]. The method used in previous work [5] was followed to generate the first predictor pair, termed RS, which was formed by rectifying the speech stimulus (and the stimulus with opposite sign).

**Gammatone spectrogram predictor (GT).** Incoming sounds undergo several stages of non-linear processing in the human ear and cochlea. The gammatone filterbank is a simple approximation of this system [31]. A gammatone filterbank consisting of 31 filters from 80-8000 Hz with 1 equivalent rectangular bandwidth (ERB) spacing was applied to the stimulus pair. The resulting amplitude spectra were averaged over all bands to generate the second predictor pair, which was termed GT. The Auditory Modeling Toolbox (AMT) version 1.1.0 [32] (function auditoryfilterbank with default parameters) was used.

**Simple model without adaptation (OSS).** The next predictor pair, termed OSS, was generated using the auditory model provided in [24], which is based on the model in [17]. The implementation in AMT (function osses2021) was used, and the generated predictors are henceforth referred to as OSS after the first author of the relevant publication [24]. This model consists of an initial headphone and outer ear pre-filter (stage 1), a gammatone filterbank (stage 2), and an approximation of inner hair cell transduction that includes rectification followed by lowpass filtering (stage 3). The next stage of the model consists of adaptation loops (stage 4), which approximate the adaptation properties of the auditory nerve. The initial prefilter was omitted since it is not required for stimuli presented with insert earphones. The adaptation stage was also omitted for this version of the model. Therefore, only stages 2 and 3 were used, and the resulting signals with 31 center frequencies (similar to GT) were averaged together to form the predictor pair.

**Simple model with adaptation (OSSA).** The adaptation loops (stage 4) of the previous auditory model [24] were now included (i.e., stages 2, 3 and 4 were used). The 31 channel output from the adaptation loops were averaged together to generate the pair of predictors. These predictors are henceforth referred to as OSSA (OSS + Adaptation).

**Complex model (ZIL).** Finally, a more complex auditory model [20] was used to generate predictors, which are henceforth referred to as ZIL after first author of the relevant publication [20]. This model has been recently used to estimate subcortical TRFs [21] and consists of several stages approximating non-linear cochlear filters, inner and outer hair cell properties, auditory nerve synapses, and adaptation. The implementation in the Python cochlea package [33] was used with 43 auditory nerve fibers with high spontaneous firing rates and center frequencies logarithmically spaced between 125 Hz and 16 kHz, in line with previous work [21]. To speed up computation, an approximation of the power-law adaptation was used [21]. The outputs of this model are the mean firing rates of the auditory nerves, which were averaged to form the final predictor pair.

## Temporal response function estimation

TRFs were fit for each predictor using the frequency domain method outlined in previous studies [5, 13] and shown in Eq (1).

$$TRF \quad = \mathcal{F}^{-1}\left\{ \frac{\sum_{i=1}^{N} w_i \mathcal{F}\{x_i\}^* \mathcal{F}\{y_i\}}{\sum_{i=1}^{N} \frac{1}{N} \mathcal{F}\{x_i\}^* \mathcal{F}\{x_i\}} \right\} \tag{1}$$

Here, $\mathcal{F}$ denotes the Fourier transform, $N$ is the number of trials, $x_i$, $y_i$ and $w_i$ are the predictor, EEG signal and weight for trial $i$, and $*$ denotes the complex conjugate. The trial weights $w_i$ were set to be the reciprocal of the variance of the EEG data of trial $i$ normalized to sum to 1 across trials. In line with prior work [13], this was done to down-weight noisy (high variance) EEG trials. This frequency domain method results in TRFs with lags from $-T/2$ to $T/2$ where $T$ is the data length.

Two TRFs were estimated separately for each predictor pair, and then averaged together. These TRFs were then bandpass filtered between 30-1000 Hz using a delay compensated FIR filter and then smoothed using a Hamming window of width 2 ms. The smoothing step was necessary since this unregularized TRF approach resulted in noisy estimates for the OSS and OSSA models (see Discussion). Although smoothing could obscure early subcortical peaks, there were no clear early peaks detected visually in the TRFs without smoothing. Given that incorporating smoothing led to more distinct wave V peaks and cleaner TRFs (less noise in the baseline period), it was used for all further TRF analysis. The TRF segment from -10 to 30 ms was extracted for further analysis. Finally, the baseline activity (mean of the TRF segment from -10 to 0 ms) was subtracted from each TRF.

To investigate the effect of data length, TRFs were estimated on a consecutively increasing number of trials (i.e, 2, 3, . . ., 8 trials, corresponding to 8, 12, . . ., 32 minutes of data) in the order that they were presented in the experiment. This simulates TRF estimation as if the experiment had been terminated after a few trials. For each data length, a leave-one-out cross-validation approach was followed, with one trial being used as test data to estimate model fits and the other trials being used to fit the TRF. The TRFs for each cross-validation fold were averaged together to form the final TRF for that data length. This resulted in 7 TRFs for each predictor that allowed for quantifying the improvement of TRF estimation with increasing data length.

## Performance metrics and statistical tests

The goodness of fits of the TRF models were evaluated using prediction correlations. The average TRF across positive and negative predictors fit on the training dataset was used to predict the EEG signal of the test trial by convolving it with the appropriate predictors, and subsequently the Pearson correlation between the predicted EEG and the actual EEG signal was calculated. The correlations across all cross-validation folds were averaged together to form an estimate of the model fit. To estimate the noise floor, a null model was formed by averaging the prediction correlations from TRFs that were fit on circularly shifted predictors (shifts of 30, 60 and 90 seconds), similar to typical null models used in prior work with cortical TRFs [12]. This method preserves the temporal structure of the stimulus, while destroying the alignment between the stimulus and the EEG, resulting in an estimate of the noise floor. The same leave-one-out cross-validation approach at each data length was followed for the null models.

The most prominent feature of ABR TRFs is the wave V peak that occurs around 5-10 ms [5, 14, 15]. The amplitude of this wave V peak was used as the primary metric for comparing

TRFs from each predictor type. The SNR of the wave V peak was computed, similar to prior work [5]. First, the TRF peak between 5-10 ms was automatically detected, and the power in a 5 ms window around the peak was computed as a measure of the signal power $S$. Next, the noise power $N$ was estimated as the average TRF power in 5 ms windows in the range -500 to -20 ms. Finally, the wave V SNR was computed as $SNR = 10\log_{10}(S/N)$. Since the signal power cannot theoretically be lower than the noise floor (i.e., 0 dB SNR), negative SNRs were assumed to be meaningless and were set to be 0 dB. The threshold for detecting meaningful wave V peaks was considered to be 3 dB (signal power is twice the noise power). This threshold of 3 dB, though arbitrary, has the intuitive meaning of the signal power being twice the noise power. Indeed, individual TRFs with more than 3 dB SNR showed visually distinct wave V peaks, confirming that this value was a reasonable threshold for wave V peak detection.

The amplitudes and latencies of the TRF wave V for each predictor for each participant were also extracted. The consistency of individual wave V was investigated using correlations of wave V amplitudes and latencies across the different predictors.

Statistical analysis was performed using non-parametric tests since the wave V SNRs have a skewed distribution with some TRFs having 0 dB SNRs (i.e., no clear wave V peaks for RS predictor). Non-parametric small sample two-tailed Wilcoxon signed rank tests with Holm Bonferroni multiple comparisons correction were used to test pairwise differences in wave V SNR across predictors. Two participants were excluded from the statistical tests since they did not have data for the full 32 minutes. The group medians, test statistics (rank sums above zero) and p-values are reported.

## Results

### Subcortical TRFs for predictors derived from auditory models

A comparison of the computational time required to generate each predictor and their correlations with the simplest (RS) and the most complex (ZIL) models are provided in Table 1. The computations were performed on an AMD Ryzen 7 PRO 5850U 1.9 GHz CPU with 32 GB RAM. Note that even the approximate ZIL model is more than 50 times slower than the others.

The grand average TRFs for the five predictors over all 24 participants are shown in Fig 2 on the left panel. The TRFs for all predictors show clear wave V peaks. The wave V peak latency slightly varies across the predictor types, even after removing lags arising from the models themselves by shifting each predictor to have the maximum correlation with RS (see Discussion). The right panel shows the model fits for each predictor as well as the corresponding null model fits. Both OSSA and ZIL show an improvement in model fits over the other 3 models. All the individual TRFs are also shown in Fig 3, highlighting the consistent wave V peak across all participants.

### Interaction of data length and predictor type on subcortical TRFs

The amount of data required for estimating the subcortical TRFs was investigated by fitting TRFs on an increasing number of 4 minute trials. Two metrics, the model fit and the wave V SNR, were used to compare TRFs across predictors and data lengths as shown in Fig 4. Almost all participants reached above zero prediction correlation and above 3 dB wave V SNR with 12 minutes of data for the OSSA and ZIL models. Two trends can be observed in Fig 4; 1) models with filterbanks (GT, OSS) produce wave V estimates with higher SNR compared to RS, 2) models with adaptation and level dependency (OSSA, ZIL) have higher wave V SNR compared to models with filterbanks. Interestingly, wave V SNR and prediction correlation of the simpler OSSA model was comparable to the more complex ZIL model. Statistical tests were performed

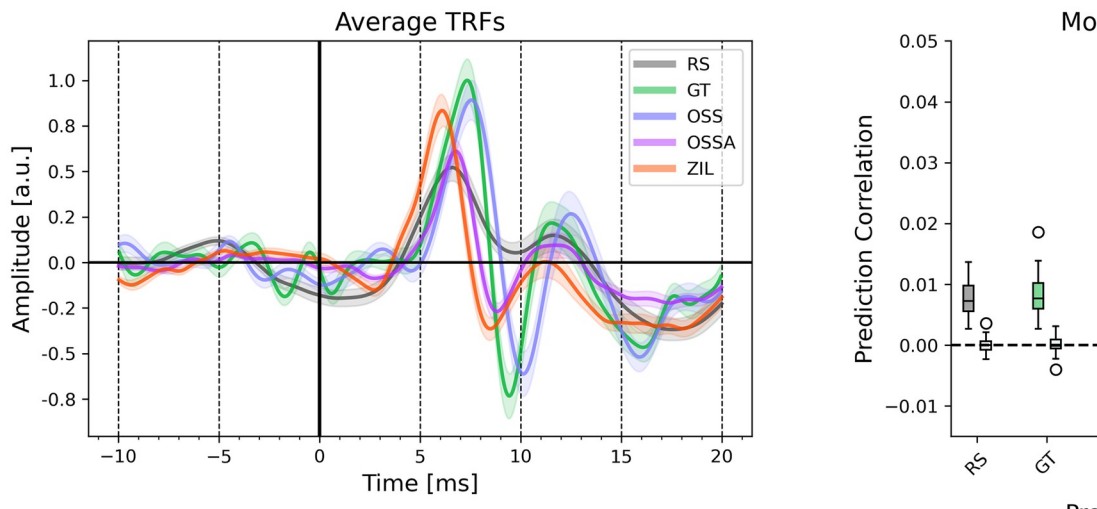

**Fig 2. Estimated TRFs for each predictor. Left**: The mean TRFs across 24 participants are shown. The standard error of the mean is also shown as the lighter shaded region. Clear wave V peaks are seen for all TRFs. **Right**: The model fit prediction correlations across participants are shown for each predictor. The prediction correlations for the null models are shown by the lighter colored boxplots next to each predictor. Outlier datapoints are marked using circles. The OSSA and ZIL models have noticeably better model fits compared to the other models.

on the wave V SNR for 32 minutes of data using pairwise non-parametric two-tailed Wilcoxon signed rank tests with Holm-Bonferroni correction. RS had significantly lower wave V SNRs (median 6.55 dB) than all other predictors (GT vs. RS $T = 32$, $p = 0.005$; OSS vs. RS $T = 49$, $p = 0.031$; OSSA vs. RS $T = 0$, $p < 0.001$; ZIL vs. RS $T = 0$, $p < 0.001$). Wave V SNRs for GT (median 9.38 dB) was not significantly different than for OSS (median 9.14 dB) ($T = 59$, $p = 0.055$). The wave V SNRs for OSSA (median 13.58 dB) and ZIL (median 13.99 dB) were larger than for OSS (OSSA vs. OSS $T = 0$, $p < 0.001$; ZIL vs. OSS $T = 2$, $p < 0.001$) or GT (OSSA vs. GT $T = 0$, $p < 0.001$; ZIL vs. OSS $T = 2$, $p < 0.001$). Critically, there was no significant difference in wave V SNRs between OSSA and ZIL ($T = 126$, $p > 0.5$), indicating that the simpler OSSA model provided comparable wave V peak amplitudes to the complex ZIL model.

### Individual amplitudes and latencies of wave V

Finally, the TRFs for the OSSA and ZIL predictors were compared as shown in Fig 5, to further investigate their similarity. The OSSA model showed a high degree of correlation with the ZIL model on a single participant level (Pearson correlation of OSSA vs. ZIL: r = 0.865 for wave V SNR, r = 0.913 for the peak latencies, r = 0.934 for the peak amplitudes, and r = 0.852 for model fits). This confirms that both models provide TRF wave V estimates that are consistent for each participant. However, the ZIL model has a shorter mean latency, also seen in Figs 2 and 3 (see Discussion). Additionally, the OSSA model seems to have slightly smaller wave V peaks than the ZIL model. Nevertheless, this correlation analysis indicates that the simpler OSSA model may provide a good trade-off between computational efficiency and reliable wave V peaks. Additionally, individual wave V SNR and model fits show moderate correlation as seen in Fig 5 (OSSA r = 0.485, ZIL r = 0.538), indicating that higher model fits may not always lead to higher wave V peaks. Therefore the appropriate metric should be considered based on whether the goal is to detect wave Vs or to evaluate model estimation quality.

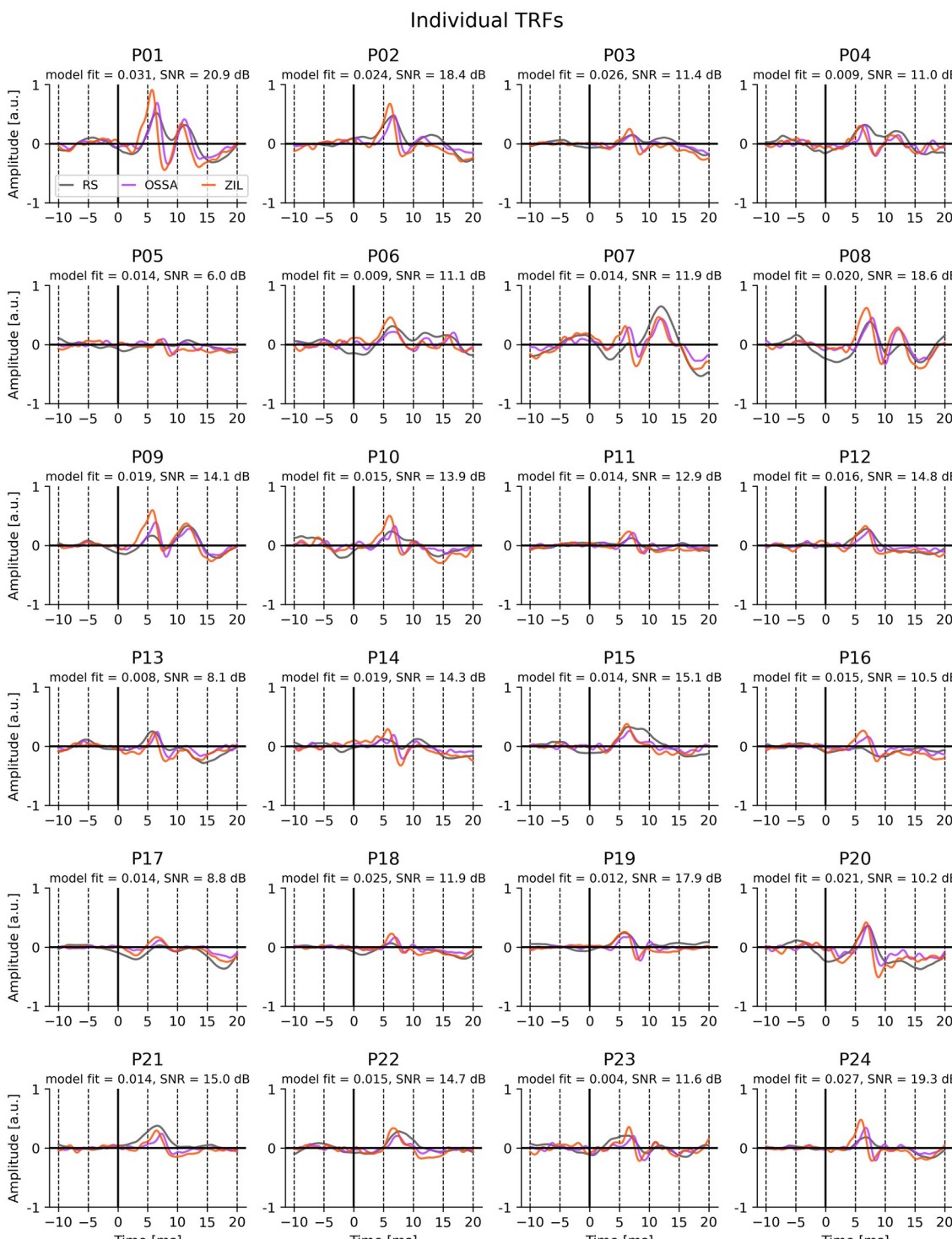

**Fig 3. Individual TRFs for all 24 participants.** For visual clarity, only TRFs for RS (gray), OSSA (magenta) and ZIL (red) predictors are shown. The model fit prediction correlation and wave V SNR for the ZIL TRF is shown above each subplot. Wave V peaks can be seen for all participants.

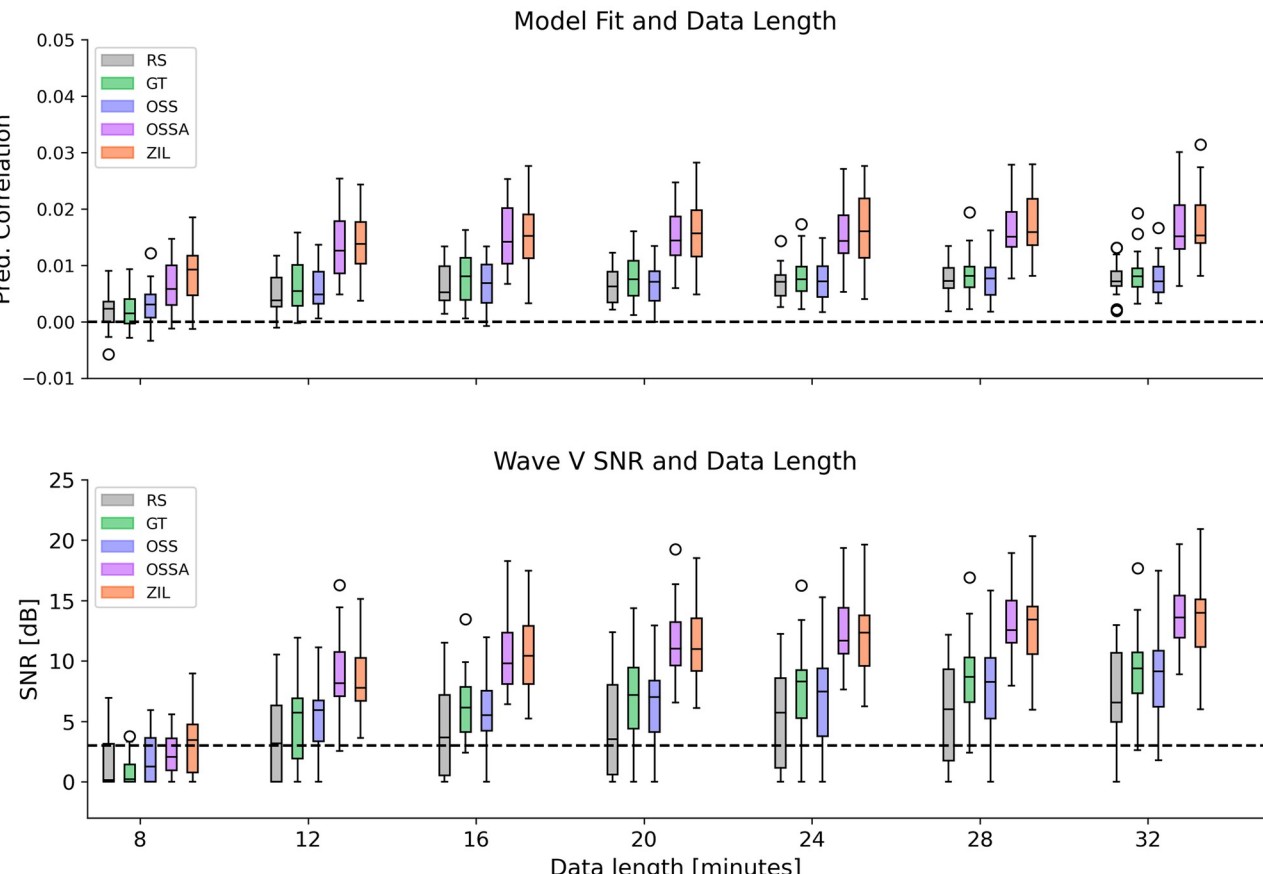

**Fig 4. Effect of predictor type and data length. Top**: Change in model fit prediction correlation with data length. Prediction correlations are shown after subtracting the corresponding null models. **Bottom**: Change in wave V SNR with data length. The threshold of 3 dB SNR is shown as a dashed line. All boxplots are shown across participants. Model fits are above zero for OSSA and ZIL for all participants after 12 minutes of data. Wave V SNRs are above 3 dB for almost all participants for OSSA and ZIL after 12 minutes of data. The wave V SNRs at 32 minutes of data for the OSSA and ZIL were significantly larger than all other predictors (see Results). Crucially, OSSA wave V SNRs were not significantly different to ZIL.

## Discussion

In this work, we compared the suitability of several predictors for estimating subcortical TRFs to continuous speech. We replicated prior work and showed that including non-linearities in the predictor using auditory models leads to improved linear TRF estimates. Our results indicate that the addition of filterbanks and adaptation stages to the predictor models greatly improves estimation of wave V in the TRFs over the rectified speech predictor. Critically, we show that even simpler models may allow for robust model fits and wave V peaks using around 12 minutes of data. These simple models give TRFs that are comparable to a more complex model, even though the complex model is more than 50 times slower to compute. However, it must be noted that OSSA wave V SNRs were comparable to ZIL only after smoothing the TRFs using a 2 ms Hamming window (see Methods), perhaps because the OSSA TRFs were noisier. Other methods such as regularized regression, which is widely used for cortical TRFs [9, 34, 35], or direct estimation of TRF peaks [36] may be able to overcome this issue. Nevertheless, our correlation analysis revealed that these smoothed TRFs resulted in wave V peak amplitudes and latencies for OSSA and ZIL that were consistent across participants.

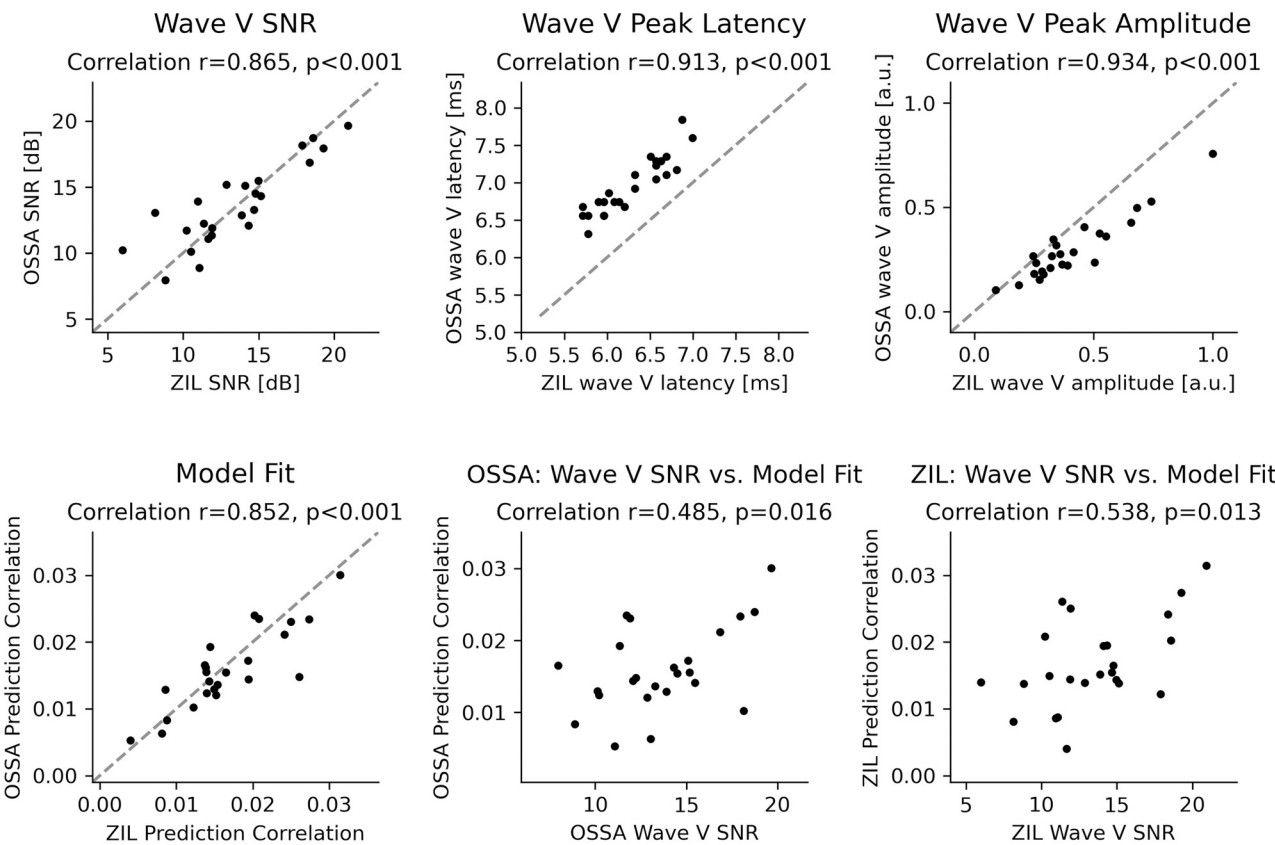

**Fig 5. Comparison of ZIL and OSSA models.** The ZIL and OSSA models are compared using several metrics and scatterplots across participants are shown. For each subplot, the Pearson correlation is also shown with corresponding p-values (Holm-Bonferroni corrected). **Top row**: Wave V peak SNR (left), latency (middle) and amplitudes (right). A high degree of correlation is seen for all three metrics. **Bottom row**: Model fits (left), model fits vs. wave V SNR for OSSA (middle) and ZIL (right). Both OSSA and ZIL models give consistent results at an individual participant level. However, the latter two subplots do not show as high a correlation between wave V SNRs and model fits for either model, indicating that it is important to look at both metrics when drawing conclusions regarding wave V peaks or TRF estimation accuracy.

The auditory models considered in this work can be categorized into three groups: rectification only (RS), models with filterbanks but without adaptation (GT and OSS), and models with adaptation (OSSA, ZIL). It should be noted that models with filterbanks provided an improvement in TRFs over rectification alone, and that models with adaptation provide the best TRFs. These results are as expected, since including these non-linearities of the peripheral auditory system in the predictor should lead to better linear TRF models.

However, it is surprising that the simpler OSSA model performs as well as the more complex ZIL model. The OSSA model is a functional model that simulates behavioral results while the ZIL model is a phenomenological model that simulates biophysical properties of the neural system [37]. Indeed the ZIL model has several stages that are absent in the OSSA model, such as adaptive filterbanks that simulate inner and outer hair cell activity, power-law adaptation, and models of the auditory synapse. However, our results indicate that perhaps such complex simulations are not necessary for estimating reliable TRF wave Vs. This does not indicate that the OSSA model simulates the auditory system as accurately as the ZIL model for other types of metrics (see [37] for a more detailed comparison of each model), but only that it may suffice for accounting for peripheral non-linearities in TRF estimation in a computationally efficient manner.

This work does not provide an exhaustive list of auditory models or predictors for estimating subcortical TRFs. We also do not directly compare the performance of the auditory models themselves (see [37]), but only evaluate their suitability to generate predictors for subcortical TRFs. Several other models (e.g., [38, 39]) could be utilized to generate predictors, although our work suggests that simple models are reliable enough to fit TRFs with clear wave V peaks.

It must be noted that although the wave V peak was used as the primary metric of performance, the conventional click ABR consists of several other morphological features [1]. The wave V peak was selected here to both be consistent with prior work [5, 15, 21], and because it was the only consistent feature that was visually detected in all participants (see Fig 3). Conventional click-ABR studies show that early peaks of the ABR are weaker with increasing stimulus rate, and that the wave V is the most consistently detected for different stimulus rates and amplitudes [2]. Therefore, these early peaks may be more difficult to detect using a continuous stimulus like speech, although one study has shown that it may be possible for some participants [21]. Future work could explore if improved predictors or TRF methods could help detect these early subcortical peaks.

TRFs using the ZIL predictor had shorter wave V peak latencies (see Figs 2 and 5), even after accounting for modelling delays by shifting the ZIL predictor to have the maximum correlation with the RS predictor. It is possible that the wave V from the ZIL model is earlier since the ZIL model better incorporates peripheral non-linearities. This may provide a predictor that is similar to intermediate signal representations in the auditory pathway near the wave V generators, which could in turn result in an earlier estimated wave V. Further investigation is needed to disentangle the effects of lags introduced by the auditory peripheral models in order to ascertain whether these latency differences are meaningful properties of the ABR.

Finally, this work only analyses subcortical responses to *speech* stimuli. Recent work indicates that complex auditory model predictors (ZIL) provide significant advantages over rectified speech when estimating subcortical TRFs for *music* [21]. Future work could investigate the suitability of simpler auditory model predictors for estimating TRFs for non-speech stimuli.

## Conclusion

This work provides a systematic comparison of predictors derived from auditory peripheral models for estimating subcortical TRFs to continuous speech. Our results indicate that simple models with filterbanks and adaptation loops may suffice to estimate reliable subcortical TRFs. Such efficient algorithms may pave the way toward the use of more ecologically relevant natural speech for investigating hearing impairment and for future assistive hearing technology.

## Acknowledgments

The authors are grateful to all participants for their participation in this study.

## Author Contributions

**Conceptualization:** Joshua P. Kulasingham, Florine L. Bachmann, Martin Enqvist, Hamish Innes-Brown, Emina Alickovic.

**Data curation:** Joshua P. Kulasingham, Florine L. Bachmann, Kasper Eskelund.

**Formal analysis:** Joshua P. Kulasingham.

**Funding acquisition:** Martin Enqvist, Hamish Innes-Brown, Emina Alickovic.

**Investigation:** Florine L. Bachmann, Kasper Eskelund, Hamish Innes-Brown, Emina Alickovic.

**Methodology:** Joshua P. Kulasingham, Florine L. Bachmann.

**Project administration:** Florine L. Bachmann, Martin Enqvist, Hamish Innes-Brown, Emina Alickovic.

**Resources:** Florine L. Bachmann.

**Software:** Joshua P. Kulasingham, Florine L. Bachmann.

**Supervision:** Martin Enqvist, Hamish Innes-Brown, Emina Alickovic.

**Validation:** Joshua P. Kulasingham.

**Visualization:** Joshua P. Kulasingham.

**Writing – original draft:** Joshua P. Kulasingham.

**Writing – review & editing:** Joshua P. Kulasingham, Florine L. Bachmann, Kasper Eskelund, Martin Enqvist, Hamish Innes-Brown, Emina Alickovic.

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
