## [Decision Letter · Decision Letter 0]

8 Nov 2023

PONE-D-23-31966Predictors for estimating subcortical EEG responses to continuous speechPLOS ONE

Dear Dr. Kulasingham,

Thank you for submitting your manuscript to PLOS ONE. After careful consideration, we feel that it has merit but does not fully meet PLOS ONE’s publication criteria as it currently stands. Therefore, we invite you to submit a revised version of the manuscript that addresses the points raised during the review process.

 As Academic Editor, I agree with both reviewers and their suggestions for addressing several points in the manuscript.

We look forward to receiving your revised manuscript.

Kind regards,

Diego A. Forero, MD; PhD

Academic Editor

PLOS ONE

Journal Requirements:

Additional Editor Comments:

I agree with both reviewers in their suggestions of several points to be addressed.

Reviewers' comments:

Reviewer's Responses to Questions

**Comments to the Author**

1. Is the manuscript technically sound, and do the data support the conclusions?

Reviewer #1: Partly

Reviewer #2: Yes

2. Has the statistical analysis been performed appropriately and rigorously? 

Reviewer #1: No

Reviewer #2: Yes

3. Have the authors made all data underlying the findings in their manuscript fully available?

Reviewer #1: No

Reviewer #2: Yes

4. Is the manuscript presented in an intelligible fashion and written in standard English?

Reviewer #1: Yes

Reviewer #2: Yes

5. Review Comments to the Author

Reviewer #1: In this paper, the authors investigated the predictors for estimating the subcortical EEG response to continuous speech. This study provides a method to model and detect the subcortical processing of continuous speech. Even though this paper has some merits and is interesting, I still feel confused about several points that need to be addressed by the authors in details.

1. The abstract has not provided numerical results that make the audiences clear after glancing the abstract.

2. Some important concepts are not clear and confusing. “Predictors” is what I concern most. TRF is recognized as predictor derived from the stimulus, but the authors also “compare predictors … for estimating TRFs”. Moreover, “rectified speech waveform as a predictor”. Why the waveform can be predictor also? It is really confusing me. It is strongly suggested to clearly define the “predictor”.

3. Some details are missing. Which ear was given stimuli? How to prove the speech segments convey the same energy, or power (72 dB SPL), as the compared methods? From the spectrogram? It is said that the electrodes were placed on the mastoids and earlobes. What’s the usage of the electrodes on earlobes? There is no information for this. How the speech stimuli were given to the subjects? In what manner? What are the parameters? What is the difference between Rectified Speech and the one in [5]? What does they look like? It is suggested to provide some figures to show the difference between RS, GT, OSS, OSSA, and ZIL.

4. More details should be provided for the model fit. Since this is an important metric for evaluation. Also, how these aforementioned methods derived from the auditory model should be clarified.

5. I am wondering the cut-off frequency of highpass filter is appropriate or not. First, after filtering, is the speech sound naturally? Second, in PTA, we know that the 500 Hz was also tested for the human being. Also, in tone-burst ABR, 500 Hz could also induce highly recognized ABR signal. It is appropriate or not to use 1kHz highpass filter to process the audio should be investigated.

6. Grounded metal box can eliminate the electromagnetic noise. But how can this avoid stimulus artifact? Please provide more details about this. For click, stimulus polarity alternation can avoid the stimulus artifact, how can speech stimulus make it, since from the results I can hardly see the stimulus artifacts.

7. The abbreviation should be rigorous. Simple model without adaptation (OSS). I don’t think OSS could be the abbr. of this phrase. So do OSSA, ZIL.

8. For figure 1, the authors should provide an explanation for the markers in the figures, like what the circles mean? Also, they should give the SD of the TRFs, and so does figure 2.

9. For figure 3, the statistical analyses should be marked out in the figure.

10. For the whole manuscript, there are discussing wave V, I think indeed they are studying ABR? If so, why the put the results for 30ms? If not, what other indexes they used?

This works made some efforts on this field but need to provide more comprehensive and detailed results for supporting their statements.

Reviewer #2: The submitted article compares 5 computational methods for estimating the subcortical TRF based on Wave V of the ABR from EEG during a passive listening task. Each model is sufficiently described in detail, and the results/conclusions indicate that while the most complex method based on a well-known auditory-nerve model (Zilany et al. 2009) was the best predictor of the TRF, it was computationally much higher than a simpler model with adaptation (OSSA), which also performed well. This leads the author to suggest that practical consideration, for example in assistive listening devices, would perhaps benefit from this computational-performance trade off.

The article is free of any glaring issues, though i have a few questions/comments that could be helpful to the reader.

1) the choice of 3 dB threshold for detecting a meaningful wave V is seemingly arbitrary and not discussed. how might the results change (if at all) if this choice were more liberal or conservative?

2) it looks like a few subjects (P01, P07, and P08) have very strong TRFs relative to the others in Figure 2. I may have missed it, but how was this taken into account when performing a grand average? That is, were their relative reponses between computational methods in any way over represented in the grand average, or was their a normalization process used?

3) In figure 3, the rightmost panel of the top row (prediction correlations) should be identical to the boxplots in Figure 1, both of which considered the full 32 minutes? It appears that there are slight differences, especially in the outliers, so this was confusing to me.

4) were tests of statistical significance run on the correlations presented in Fig 4?

5) check the references - some do not contain the publication year, like Picton and Zilany et al.

6. PLOS authors have the option to publish the peer review history of their article (what does this mean?). If published, this will include your full peer review and any attached files.

Reviewer #1: No

Reviewer #2: No

---

## [Author Response · Author response to Decision Letter 0]

12 Dec 2023

We have responded to all reviewer and editor comments in our response to reviewers pdf document.

---

## [Decision Letter · Decision Letter 1]

15 Jan 2024

Predictors for estimating subcortical EEG responses to continuous speech

PONE-D-23-31966R1

Dear Dr. Kulasingham,

We’re pleased to inform you that your manuscript has been judged scientifically suitable for publication and will be formally accepted for publication once it meets all outstanding technical requirements.

Kind regards,

Diego A. Forero, MD; PhD

Academic Editor

PLOS ONE

Additional Editor Comments:

Both reviewers recommend the acceptance of the revised manuscript.

Reviewers' comments:

Reviewer's Responses to Questions

**Comments to the Author**

1. If the authors have adequately addressed your comments raised in a previous round of review and you feel that this manuscript is now acceptable for publication, you may indicate that here to bypass the “Comments to the Author” section, enter your conflict of interest statement in the “Confidential to Editor” section, and submit your "Accept" recommendation.

Reviewer #1: All comments have been addressed

Reviewer #2: All comments have been addressed

2. Is the manuscript technically sound, and do the data support the conclusions?

Reviewer #1: Yes

Reviewer #2: Yes

3. Has the statistical analysis been performed appropriately and rigorously? 

Reviewer #1: N/A

Reviewer #2: Yes

4. Have the authors made all data underlying the findings in their manuscript fully available?

Reviewer #1: Yes

Reviewer #2: Yes

5. Is the manuscript presented in an intelligible fashion and written in standard English?

Reviewer #1: Yes

Reviewer #2: Yes

6. Review Comments to the Author

Reviewer #1: The authors have addressed the question i am confused. And i think the manuscript is now ready to be accepted.

Reviewer #2: (No Response)

7. PLOS authors have the option to publish the peer review history of their article (what does this mean?). If published, this will include your full peer review and any attached files.

Reviewer #1: No

Reviewer #2: No

---

## [Editor Report · Acceptance letter]

30 Jan 2024

PONE-D-23-31966R1 

PLOS ONE

Dear Dr. Kulasingham, 

I'm pleased to inform you that your manuscript has been deemed suitable for publication in PLOS ONE. Congratulations! Your manuscript is now being handed over to our production team.

Kind regards, 

on behalf of

Dr. Diego A. Forero 

Academic Editor

PLOS ONE